# Generative Motion Stylization of Cross-structure Characters within Canonical Motion Space

Jiaxu Zhang
Wuhan University
Wuhan, China
zjiaxu@whu.edu.cn

Xin Chen
Tencent PCG
Shanghai, China
chenxin2@shanghaitech.edu.cn

Gang Yu
Tencent PCG
Shanghai, China
skicy@outlook.com

Zhigang Tu*
Wuhan University
Wuhan, China
tuzhigang@whu.edu.cn

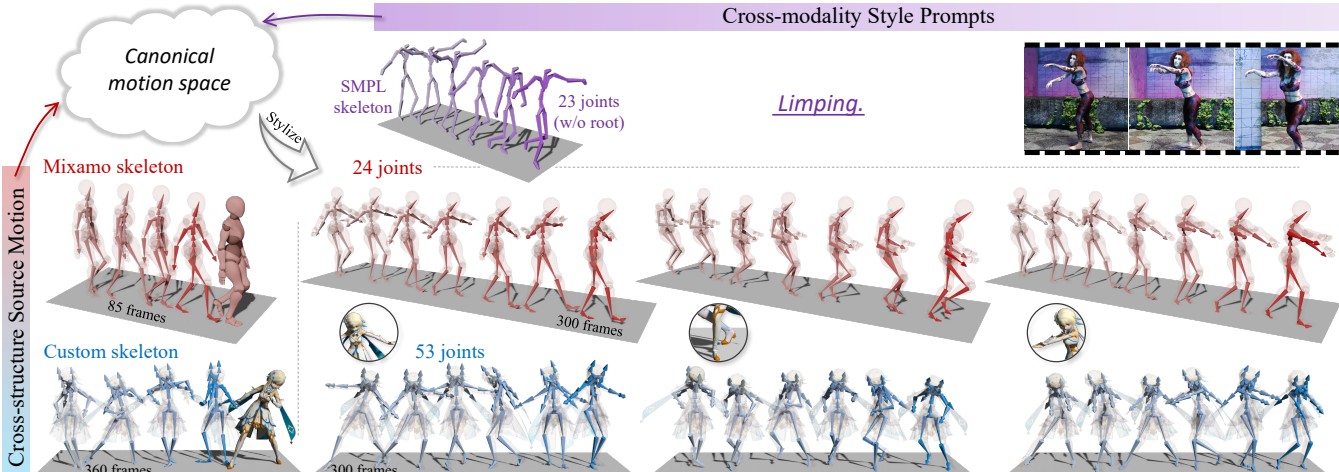

**Figure 1: We present Motion𝕊, a generative motion stylization pipeline for synthesizing diverse and stylized motion on cross-structure source motion with usage of cross-modality style prompts.**

## ABSTRACT

Stylized motion breathes life into characters. However, the fixed skeleton structure and style representation hinder existing data-driven motion synthesis methods from generating stylized motion for various characters. In this work, we propose a generative motion stylization pipeline, named Motion𝕊, for synthesizing diverse and stylized motion on cross-structure characters using cross-modality style prompts. Our key insight is to embed motion style into a cross-modality latent space and perceive the cross-structure skeleton topologies, allowing for motion stylization within a canonical motion space. Specifically, the large-scale Contrastive-Language-Image-Pre-training (CLIP) model is leveraged to construct the cross-modality latent space, enabling flexible style representation within

it. Additionally, two topology-encoded tokens are learned to capture the canonical and specific skeleton topologies, facilitating cross-structure topology shifting. Subsequently, the topology-shifted stylization diffusion is designed to generate motion content for the particular skeleton and stylize it in the shifted canonical motion space using multi-modality style descriptions. Through an extensive set of examples, we demonstrate the flexibility and generalizability of our pipeline across various characters and style descriptions. Qualitative and quantitative comparisons show the superiority of our pipeline over state-of-the-arts, consistently delivering high-quality stylized motion across a broad spectrum of skeletal structures.

## CCS CONCEPTS

• **Computing methodologies** → **Motion processing**; *Computer vision tasks*.

## KEYWORDS

Motion stylization, motion generation, cross-structure, cross-modality

**ACM Reference Format:**
Jiaxu Zhang, Xin Chen, Gang Yu, and Zhigang Tu. 2024. Generative Motion Stylization of Cross-structure Characters within Canonical Motion Space. In *Proceedings of the 32nd ACM International Conference on Multimedia (MM'24), October 28-November 1, 2024, Melbourne, Australia.* ACM, New York, NY, USA, 9 pages. https://doi.org/10.1145/3664647.3680864

*Corresponding author

# 1 INTRODUCTION

*"Style is the answer to everything."* – Charles Bukowski

Style serves as a pivotal element in animations, mirroring diverse facets of a character, including personality, habits, emotions, and intentions. Therefore, the expressiveness of motion style plays a crucial role in motion synthesis and enriches storytelling in computer animation. However, the complexity and time-consuming nature of the traditional motion stylization process presents significant barriers to entry [19]. In recent years, learning-based motion synthesis methods have emerged in the community [10, 38]. Still, they are often constrained to a fixed skeleton structure, generating motion content without expressing the nuances of motion styles. Thus, they fall short of synthesizing stylized motion for a wide variety of characters.

By delving deeper into the underlying reasons, we have identified two primary challenges impeding the development of a comprehensive stylized motion synthesis. The first challenge involves expressing motion style on cross-structure skeletons. Existing methods [2, 9, 27] for motion style transfer oversimplify this obstacle by assuming a fixed skeletal structure between the source and target characters, thereby neglecting the intricacies of the characters' skeletal topologies. This inherent assumption significantly limits their applicability for transferring motion style among characters with disparate skeletal structures. Moreover, the predominant focus of existing stylized motion datasets [26, 39] is on standard human skeletons, making it difficult for learning-based methods to stylize non-standard skeletal motions. Consequently, perceiving the skeleton topology and effectively expressing motion style on cross-structure skeletons emerge as a critical task.

The second challenge pertains to the utilization of cross-modality style representations. Existing motion generation methods [13, 21] have explored various conditions to control the content of the generated motion. However, these methods struggle to effectively control the motion style through these multi-modality conditions, resulting in the generated motion often perceived as dull and monotonous. Additionally, the previous motion style transfer methods [33, 34] are constrained to using motion sequences or category labels as the style description, rendering flexible style control impossible. Thus, incorporating cross-modality style representations for more flexible and user-friendly motion-style editing stands out as another significant task.

To address these challenges, we introduce a new generative motion stylization pipeline, MotionS, capable of synthesizing diverse motion for a wide range of characters and stylizing it using motion sequences, text, images, or videos as style descriptions, as Figure 1 shows. In MotionS, two novel techniques, *i.e.* cross-modality style embedding and cross-structure topology shifting, are explored to construct a canonical motion space. In this space, motion content from various skeletons can be stylized using the extracted style embedding by adjusting the mean and variance of the generated motion features. With these two key techniques, the topology-shifted stylization diffusion is presented to synthesize diverse and stylized motion, establishing a more accessible way of animation creation.

Cross-modality style embedding is to embed style descriptions in various formats into the canonical motion space. Drawing inspiration from GestureDiffuCLIP [5], which employs CLIP latents [32] for the automatic creation of stylized gestures, we align the motion

features of the SMPL skeleton [24] with CLIP latents and construct a shared space that represents motion style as adaptive mean and variance applied to the motion features. The multi-modality style embedding in the shared latent space enables flexible style descriptions for motion stylization in our MotionS.

Cross-structure topology shifting aims to transfer motion features between canonical and specific motion spaces. To achieve this, we utilize two learnable topology-encoded tokens (TET) for capturing the canonical and specific skeleton topologies. Each TET is followed by a graph convolutional layer (GCL) with a pre-defined adjacency matrix, serving as a topology prior of the skeletal structure. Subsequently, the motion space is shifted using the cross-attention mechanism. This topology shifting strategy enables cross-structure motion stylization in our MotionS.

In response to the fact posed by the scarcity motion sequences of arbitrary skeletal structure, we designed our MotionS based on the SinMDM [31]. SinMDM requires only one motion sequence to learn the motion content and can synthesize variable-length motion sequences that retain the core motion elements of the single input. Remarkably, distinct from SinMDM, the focus of this work is on the stylization of cross-structure motion and the utilization of cross-modality style representations.

We evaluate MotionS across a variety of skeletal-rigged characters and style descriptions. Both qualitative and quantitative results demonstrate the effectiveness of MotionS in perceiving skeleton topologies, constructing the canonical motion space, and synthesizing stylized motion. In comparison with state-of-the-art methods, MotionS exhibits superior stylized motion results in terms of content preservation, style fidelity, and stylized motion diversity.

Our contributions are listed below:

- We propose MotionS, a generative motion stylization pipeline that, to our knowledge, marks the pioneering attempt to synthesize cross-structure diverse and stylized motion by utilizing cross-modality style descriptions.
- We exploit two novel techniques in MotionS, *i.e.*, cross-modality style embedding and cross-structure topology shifting. These techniques collaborate to construct a canonical motion space, enabling skeletal topology perception and flexible style representations for motion stylization.
- Extensive experiments demonstrate the effectiveness and generalizability of our MotionS, which achieves higher motion quality and style diversity in comparison with the prior learning-based methods.

# 2 RELATED WORK

***Motion stylization*** has been a longstanding challenge within the realm of computer animation [3, 7, 17, 42]. Recently, deep learning-based approaches [11, 15, 16, 20, 27] started sparkling in the community. For example, Xia et al. [39] proposed an online learning algorithm to capture complex relationships between pairwise motion styles. Yumer and Mitra [40] represented motion style in the frequency domain, extracting style features without the need for spatial matching. Smith et al. [33] presented a neural network-based style transfer model, enabling the adjustment of output motion in a latent space. These methods highlight the research trend of achieving flexibility and diversity in style control. However, they are limited to representing labeled or sample-based motion styles, lacking support for multi-modality style descriptions.

Moving forward, Aberman et al. [2] introduced a data-driven framework encoding motion into two latent spaces for content and style, enabling the extraction of unseen styles from videos. Nevertheless, this approach requires estimating skeletal motion from the video first and cannot directly extract style features from ordinary videos. Additionally, Jang et al. [19] introduced a Motion Puzzle framework capable of controlling the motion style of individual human body parts. Ao et al. [5] leveraged the power of the CLIP model to synthesize co-speech gestures with flexible style control. Jang et al. [20] proposed an online motion characterization framework that can transfer both the motion style and body proportions of characters. However, the skeleton structure in these methods remains fixed and they cannot handle various characters with different skeletons. In line with the current research trend, our work explores previously uncharted territory: motion stylization on cross-structure characters using cross-modality style prompts. Distinct from previous studies, our proposed pipeline is generative and possesses the ability to perceive the skeletal topology. It can flexibly generate diverse stylized motion based on style features extracted from motion sequences, text, images, or videos.

*Motion generation* has emerged as a central focus of animation creation in recent years, propelled by advancements in generative deep models [12, 14]. Recent cutting-edge methods often explore various conditions for human motion generation, which include but are not limited to action labels [13, 29], audio [4], and text [38]. For example, Tevet et al. [37] introduced the Motion Diffusion Model (MDM), aiming for natural and expressive human motion generation from text prompts. Building upon MDM, Chen et al. [8] employed a Variational Auto-encoder (VAE) to enhance motion representation, and Zhang et al. [43] integrated a retrieval mechanism to refine the denoising process. Additionally, Zhang et al. [41] proposed a text-driven animation pipeline (TapMo) for generating motion in a broad spectrum of skeleton-free characters. However, these methods typically require large amounts of data for training and overlook motion style in the generation process. In our work, recognizing the scarcity of motion data for non-standard skeletons, we draw inspiration from SinMDM [31] to construct a generative model learning from a single motion sequence. More importantly, our pipeline can control the style of cross-structure skeletal motion through multi-modality style descriptions in a learned canonical motion space.

## 3 METHOD

As illustrated in Figure 2, our Motion$\mathbb{S}$ pipeline consists of two parts, *i.e.*, a style encoder $\mathcal{E}(\cdot)$ and a diffusion model $\mathcal{F}(\cdot)$. Given a style prompt $p$ in an arbitrary representation, the style encoder extracts the style feature $f_p$ within a cross-modality latent space. Next, the style feature is input into a linear mapping layer to derive the style embedding, represented by $\mu$ and $\sigma$. These two tensors correspond to the mean and standard deviation of motion features in the canonical motion space, respectively. Let $\theta_{\mathcal{E}}^M$, $\theta_{\mathcal{E}}^T$, and $\theta_{\mathcal{E}}^I$ represent the parameters of the motion, text, and image style encoders, respectively. This process is formulated as:

$$\mathcal{E} : (p; \theta_{\mathcal{E}}^M, \theta_{\mathcal{E}}^I, \theta_{\mathcal{E}}^T) \mapsto (f_p, \mu, \sigma). \tag{1}$$

Subsequently, the diffusion model takes the noised motion $x_s$ and the step $s$ as inputs to generate the stylized motion $\hat{x}_0^p$ through $1,000$

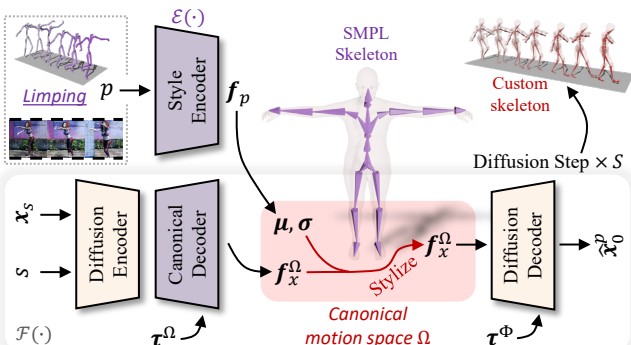

**Figure 2: An overview of the Motion$\mathbb{S}$ pipeline. Motion$\mathbb{S}$ takes multi-modality prompts $p$ as style descriptions, generates diverse motion $\hat{x}_0^p$ for specific skeletal structures through the diffusion denoising process, and performs the motion stylization in a canonical motion space $\Omega$.**

denoising steps. All motions in this work are represented using the 6D rotation features proposed by Zhou et al. [44]. Throughout this process, the style feature is equipped to provide fine-grained dynamic information about the motion style. The style embedding influences the motion feature in canonical space using the AdaIN strategy [18], controlling the style property. Notably, two TETs, *i.e.*, $\tau^{\Omega}$ and $\tau^{\Phi}$, are employed to shift the motion feature space between canonical $\Omega$ and specific $\Phi$ settings. This is achieved through the cross-attention mechanism used in the canonical decoder and the diffusion decoder, allowing the model to perceive the skeletal topologies and control motion style on cross-structure skeletons. This inverse diffuse process is formulated as:

$$\mathcal{F} : (x_s, s, f_p, \mu, \sigma; \theta_{\mathcal{F}}) \mapsto (\hat{x}_0^p), \tag{2}$$

where $\theta_{\mathcal{F}}$ denote the learnable parameter of the diffusion model. Following SinMDM [31], the diffusion encoder is constructed as a QnA-based network [6] to learn motion contents from a single sequence effectively. For the global translation in the skeleton motion, we employ an independent three-layer convolutional network in the diffusion steps to generate it. For simplification, this aspect is ignored in the following discussion.

### 3.1 Cross-modality Style Embedding

Given that the existing human motion dataset is commonly represented using the SMPL model [25], we use the SMPL motion $m$ as style prompts in motion format. Accordingly, the topology of the SMPL skeleton is utilized to construct the canonical motion space, which comprises $N^{\Omega}$ skeleton joints.

As depicted in Figure 3, drawing inspiration from MotionCLIP [36], we design a transformer-based motion encoder to embed $m$ into a style feature $f_p^M \in \mathbb{R}^{1 \times C}$ and a content feature $f_m \in \mathbb{R}^{T \times C}$, where $T$ is the time dimension and $C$ is the channel. The style embedding $\mu \in \mathbb{R}^C$ and $\sigma \in \mathbb{R}^C$ are then obtained by mapping $f_p^M$ through a linear layer. Next, we repeat $f_p^M$ along the temporal dimension and concatenate it with $f_m$ to construct the input motion feature of $2 \times T \times C$ dimensions for the canonical decoder. Subsequently, the transformer-based canonical decoder utilizes the input motion feature as the *Key* and *Value* of the cross-attention mechanism. Meanwhile, it takes the GCL refined canonical TET $\tau^{\Omega} \in \mathbb{R}^{N^{\Omega} \times C}$ as the *Query* to map the motion feature from $2 \times T \times C$ to $N^{\Omega} \times T \times C$ dimensions. At this point, the motion feature and

the style feature are all expressed in a canonical motion space. This process is formulated as:

$$\mathcal{D} : \left( \mathcal{E}(m; \theta_{\mathcal{E}}^M), \tau^\Omega; \theta_{\mathcal{D}} \right) \mapsto (f_m^\Omega, \boldsymbol{\mu}, \boldsymbol{\sigma}), \tag{3}$$

where $\mathcal{D}$ and $\theta_{\mathcal{D}}$ denote the canonical decoder and its learnable parameter, respectively. Finally, the AdaIN and a linear layer are applied to reconstruct the input motion $\hat{m}$ using $f_m^\Omega$, $\boldsymbol{\mu}$, and $\boldsymbol{\sigma}$ from the canonical motion space.

***Training.*** The training strategy of our cross-modality style embedding is similar to that of MotionCLIP, and it serves as a pre-training stage for our Motion$\mathbb{S}$. Specifically, we utilize L2 losses for the reconstruction of joint rotations $m$, positions $n$, and velocities $v$, expressed as:

$$\mathcal{L}_{rec} = \|m - \hat{m}\|_2^2 + \|n - \hat{n}\|_2^2 + \|v - \hat{v}\|_2^2. \tag{4}$$

The alignment of the style feature and the CLIP latent space is achieved through the supervision of cosine similarity losses on the triplet of the image, text, and motion style features:

$$\mathcal{L}_{sim} = 2 - cos(f_p^T, f_p^M) - cos(f_p^I, f_p^M). \tag{5}$$

With these two losses, the style encoder and the canonical decoder can be trained by:

$$\min_{\theta_{\mathcal{E}}^M, \theta_{\mathcal{D}}, \tau^\Omega} \mathcal{L}_{rec} + v_{sim}\mathcal{L}_{sim}, \tag{6}$$

where $v_{sim}$ is the loss balancing factor. $\theta_{\mathcal{E}}^M$, $\theta_{\mathcal{D}}$, and $\tau^\Omega$ are optimized in this training process.

## 3.2 Cross-structure Topology Shifting

To enable general motion stylization beyond the standard human skeleton, we introduce the cross-structure topology shifting strategy, which involves transferring the skeletal topology of the motion in the latent feature space. As illustrated in the right part of Figure 4, for a type of skeleton B with $N$ joints, we exploit a learnable topology-encoded token $\tau \in \mathbb{R}^{N \times C}$ to capture the structure and order of the skeleton joints. Leveraging the graph convolutional layer inspired by SAME [22], which incorporates a strict topology prior, we input the initial TET into a GCL to refine the encoded topology, thus facilitating the learning of the skeletal structure. This process can be formulated as:

$$\tau' = \widetilde{A}\tau W, \tag{7}$$

where $A \in \mathbb{R}^{N \times N}$ is the pre-defined adjacency matrix and $\widetilde{A} = D^{-1/2}AD^{-1/2}$. $D_{i,j} = \sum_j A_{i,j}$ is the the diagonal degree matrix. $W$ is the learnable weights in the linear layer.

Subsequently, the refined $\tau$ is fed into a cross-attention block as *Query*, while a motion feature $f^A$ in space A serves as *Key* and *Value* to transfer the motion space from one to another. Notably, the structural topology and joint order of these two motion spaces are distinctly different, and the TETs are optimized with the network in training. By employing our cross-structure topology shifting, the motion space can be switched between any custom skeleton structures and the SMPL-based canonical structure. This process enables motion stylization using the style embedding extracted from multi-modality style prompts through a unified AdaIN layer.

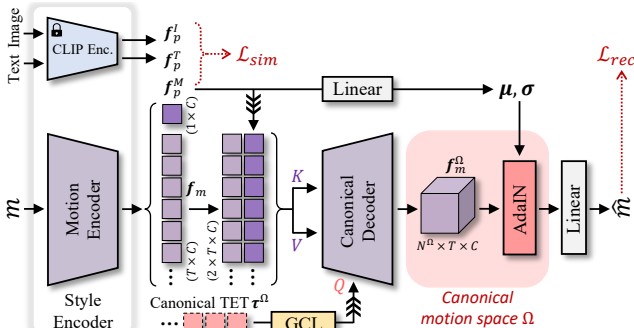

**Figure 3: The structure and training strategy of the cross-modality style embedding. We design a motion encoder to embed the motion $m$ of the standard SMPL skeleton and align its latent space with the CLIP encoders. The motion encoder and the CLIP encoders constitute the cross-modality style encoder in Motion$\mathbb{S}$. Furthermore, the canonical motion space is constructed through the canonical decoder and the learnable canonical TET.**

## 3.3 Topology-shifted Stylization Diffusion

Building on the two key techniques introduced above, we present the topology-shifted stylization diffusion (TSD) model, as illustrated in the left part of Figure 4. Our TSD stands out from existing motion stylization models for two main differences: 1) TSD is a generative model capable of synthesizing diverse motion from a single sequence, addressing the challenge of scarce motion data for custom characters. 2) TSD can stylize the generated motion for arbitrary skeleton structures using multi-modality style prompts, providing a flexible and user-friendly approach for animation creation.

Specifically, TSD comprises three network modules: the diffusion encoder, the canonical decoder, and the diffusion decoder. First, the QnA-based diffusion encoder takes the noised motion $x_s$ and the step $s$ as inputs, producing the motion content feature $f_x \in \mathbb{R}^{T' \times C}$. Simultaneously, the style encoder extracts the style feature from an SMPL motion sequence. Then, following a process similar to the cross-modality style embedding shown in Figure 3, the style feature is repeated, concatenated with the motion feature, and fed into the pre-trained canonical decoder. At this stage, the canonical decoder, integrated with the canonical TET, transfers the style feature to the canonical motion space, allowing for stylization using the style embedding. Notably, the parameters of the style encoder, canonical decoder, and canonical TET are all frozen in this process.

Subsequently, in training, the network flow is divided into two branches. In one branch, the canonical motion feature is directly fed into the transformer-based diffusion decoder to shift it into the specific motion space. In the other branch, the canonical motion feature undergoes stylization using the AdaIN layer, a process akin to cross-modality style embedding. The stylized canonical feature is then input into the diffusion decoder, yielding a stylized specific motion feature. It should be noticed that, during inference, the stylized result is obtained through the stylization branch. This unique design enables the stable training of TSD even without the ground-truth stylized motion for the specific character.

Finally, the output features from the two branches are fed into a linear layer to predict the ground-truth motion $\hat{x}_0$ and the stylized motion $\hat{x}_0^p$, respectively. Additionally, the stylized canonical motion feature $f_x^\Omega$ is also output in training for calculating the losses.

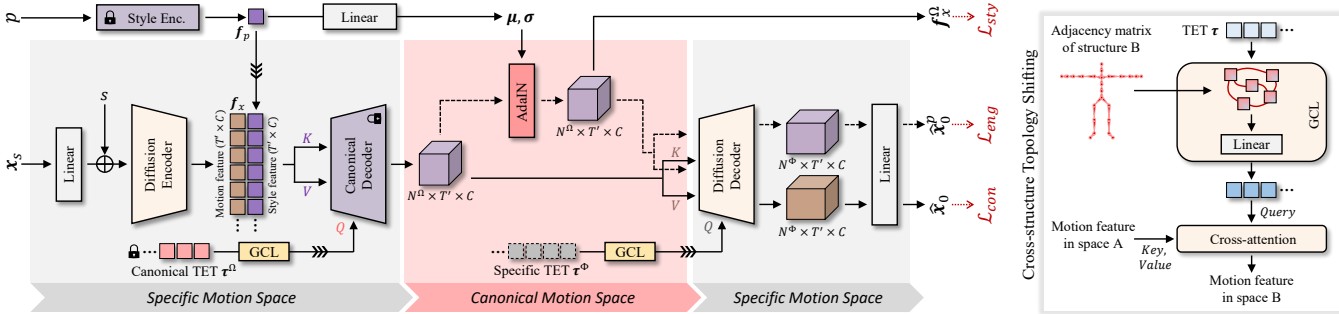

**Figure 4: Illustration of the model details. The left part is the structure and training strategy of our topology-shifted stylization diffusion (TSD) model, which generates the stylized motion across specific and canonical motion spaces. The right part is the illustration of the cross-structure topology shifting.**

*Training.* We design three losses to train the TSD in the absence of ground-truth stylized motion for specific characters, *i.e.*, the content loss $\mathcal{L}_{con}$, the energy loss $\mathcal{L}_{eng}$, and the style loss $\mathcal{L}_{sty}$.

Originating from the simple loss of DDPM [14], the content loss is formulated as:

$$\mathcal{L}_{con} := \mathbb{E}_{s\sim[1,S]}\left[\|x_0 - \hat{x}_0\|_2^2\right]. \tag{8}$$

The energy loss aims to weakly supervise the stylized results $\hat{x}_0^p$ with the style prompt motion $x^p$ using the motion energy. To address structural differences between the SMPL skeleton of the style prompt motion and the custom skeleton of the stylized motion, we group the skeleton joints into five main human body parts: torso, left arm, right arm, left leg, and right leg. Then, we average the joint rotations within each group to align the joint number and order of the two structures. Additionally, we align the temporal dimension of the stylized results and the style prompt motion using linear interpolation. The formulation of the energy loss is as follows:

$$\mathcal{L}_{eng} := \mathbb{E}_{s\sim[1,S]}\left[\left\|group\left(interp(x^p)\right) - group(\hat{x}_0^p)\right\|_2^2\right]. \tag{9}$$

The style loss aims to guarantee that the stylized canonical motion feature $f_x^\Omega$ embodies the corresponding motion style present in the style prompt motion $x^p$. To achieve this, we employ the final linear layer of the pre-trained canonical decoder to reconstruct $f_x^\Omega$ into a motion sequence compatible with the standard SMPL skeleton. Subsequently, we extract its style feature using the pre-trained style encoder. The style loss is then calculated as:

$$\mathcal{L}_{sty} = \left\|f_p - \mathcal{E}\left(linear(f_x^\Omega)\right)\right\|_2^2. \tag{10}$$

With the three losses introduced above, the TSD is trained by:

$$\min_{\theta_{\mathcal{F}}, \tau^\Phi} \mathcal{L}_{con} + v_{eng}\mathcal{L}_{eng} + v_{sty}\mathcal{L}_{sty}, \tag{11}$$

where $v_{eng}$ and $v_{sty}$ are the loss balancing factor. $\theta_{\mathcal{F}}$ and $\tau^\Phi$ are optimized in this training process.

## 4 EXPERIMENTS

*Baselines.* To the best of our knowledge, MotionS is the first pipeline capable of generating stylized motion across various skeleton structures using a range of style prompts. We utilize three reasonable baselines to evaluate the effectiveness of our MotionS.

**SinMDM** [31]. In the case of sample-based stylization, we retarget the style motion to the specific character's skeleton and employ the approach introduced in SinMDM to generate the results. For text and image-based stylization, we utilize MDM to generate the

motion samples for the SMPL skeleton. Subsequently, the stylization process aligns with the sample-based approach.

**Deep Motion Editing (DME)** [2] can utilize either motion samples or action videos as style prompts to stylize input motion sequences. However, it is limited to a fixed skeleton structure and cannot accommodate text prompts. Thus, in our comparison with DME, we follow its specific skeletal settings.

**Motion Puzzle** [19] controls the motion style of individual body parts through reference motion sequences. It only supports sample-based style prompts with a fixed skeleton structure. For the video prompts, we utilize VIBE [35] to estimate the skeleton motion and retarget it into the specific skeleton.

*Datasets.* Two public datasets, namely BABEL [30] and 100STYLE [26], are employed to train our MotionS. BABEL is used for training the cross-modality style embedding, while 100STYLE serves as the style motion prompt for training the TSD model. Further details about the datasets can be found in the supplementary material.

*Implementation details.* We implement our pipeline using Py-Torch framework[28]. In the cross-modality style embedding (refer to Section 3.1), the motion style encoder comprises a four-layer transformer-encoder, while the canonical decoder is constructed with a four-layer transformer-decoder. In TSD (refer to Section 3.3), the diffusion encoder consists of an eight-layer QnA-encoder, and the structure of the diffusion decoder mirrors that of the canonical decoder. In our pipeline, the feature channel size $C$ is set to 512. The number of the canonical joints is 23, representing the SMPL skeleton joints excluding the root joint. For video prompts, we randomly sample five frames to extract style features, which are computed as the average image features of the sampled frames. Please see the supplementary material for more details.

*Evaluation metrics.* We quantitatively evaluate our MotionS from three aspects, *i.e.*, content preservation, style fidelity and stylized motion diversity. For evaluating content preservation and style fidelity, we train TSD on two types of skeletal motions. The first consists of four SMPL motion sequences corresponding to walk, run, jump, and idle actions. The second includes two Xia et al. [39] motion sequences used in DME and Motion Puzzle, corresponding to walk and jump actions. Subsequently, we utilized the pre-trained motion style encoder to extract the motion content features and the style features of both the generated motion and the ground-truth motion. Finally, we calculated the **Fréchet Motion Distance (FMD)** [19] to measure the distance between the features of the generated motion and the ground-truth motion, reflecting the degree

**Table 1: Quantitative comparison on the SMPL source motion and the Xia et al. [39] source motion. The best results are in bold, and the second best are underlined. Note that a balance of good scores across all metrics is better than excelling in just a few.**

| Methods | SMPL Source Motion | | | | Xia Source Motion | | | |
|---|---|---|---|---|---|---|---|---|
| | Content↓ | Style↓ | Glo-D→ | Loc-D→ | Content↓ | Style↓ | Glo-D→ | Loc-D→ |
| SinMDM | 28.24 | 15.12 | 3.37 | 3.21 | 31.73 | 16.82 | 4.27 | 3.39 |
| SinMDM* | **10.24** | 31.64 | 0.81 | 0.77 | **9.12** | 37.01 | 0.74 | 0.62 |
| DME | 33.51 | 32.13 | - | - | 13.74 | **11.89** | - | - |
| Motion Puzzle | 21.95 | 17.66 | - | - | 17.47 | 12.44 | - | - |
| MotionS (CLIP) | 24.21 | 22.75 | 3.23 | 3.01 | 27.46 | 23.79 | 3.65 | 3.12 |
| MotionS (w/o TET) | 16.35 | 18.14 | 1.75 | 1.66 | 22.31 | 28.64 | 2.71 | 1.98 |
| MotionS (Ours) | 10.96 | **14.11** | **1.24** | **1.20** | 12.32 | 14.65 | **1.89** | **1.73** |

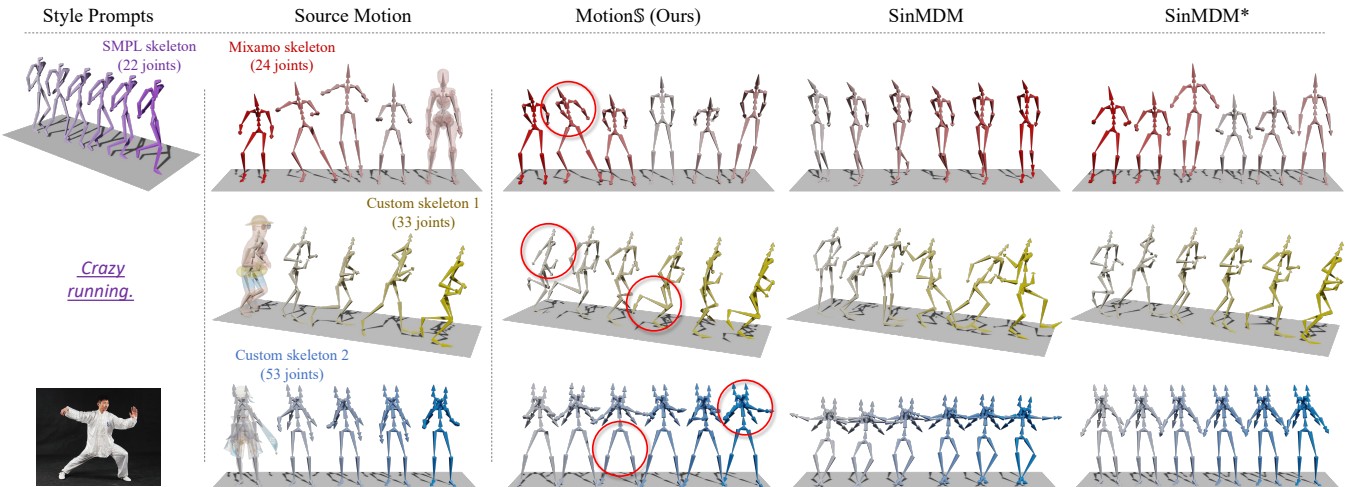

**Figure 5: Qualitative comparison with SinMDM. SinMDM\* refers to the model trained on the source motion instead of the style motion. SinMDM exhibits unstable performance in expressing both the motion content and style.**

of content preservation and style fidelity. For evaluating stylized motion diversity, we utilize the **Global Diversity (Glo-D)** and **Local Diversity (Loc-D)** metrics proposed in [23].

## 4.1 Qualitative Results

***Comparison with the baselines.*** Figure 5 illustrates the qualitative comparison between our MotionS and SinMDM. In their study, the authors trained the model on style motion and fused content motion during the inference process for stylization. However, we observed that this approach often results in generated outputs closely resembling the style motion, potentially neglecting the preservation of content motion, as shown in the fourth column of Figure 5. Thus, we additionally implemented SinMDM\* by training the model on the source motion instead of the style motion.

Our MotionS effectively handles various skeleton structures, stylizing the source motion based on key features of multi-modality style prompts. Importantly, our MotionS well preserves the motion content in the generated results. For instance, when using a walking motion with arms behind the back as the style prompt and a jumping motion as the source, our result maintains the jumping content and adjusts the skeletal arms to resemble the stylized motion. In contrast, SinMDM tends to lose the jumping content, mistakenly generating a walking motion. Furthermore, SinMDM\* fails to transfer the style to the content motion. In addition, when the skeletal structure is complex and there is a significant difference between the style prompt and source motion, both SinMDM and SinMDM\* face mode collapse, resulting in static motion sequences, as depicted in the

third row of Figure 5. In summary, MotionS inherits the strengths of SinMDM in generating diverse sequences from a single source. Additionally, MotionS is capable of learning skeletal structure and utilizing multi-modality cues for motion stylization.

Figure 6 shows the qualitative comparison of our MotionS with DME and Motion Puzzle. Due to the fixed Xia et al. [39] skeleton structure employed in their work, the style sequences were retargeted from the SMPL skeleton to their specific skeleton. We trained TSD using the two motion sequences provided in the DME's demo. Due to the uniformity of the jumping action, as shown in the first row, we added positional encoding before the diffusion decoder of TSD to ensure that the generated motion is consistent with the source. In this experiment, we noticed that DME yields impressive results when utilizing their normalized style prompts. However, when confronted with in-the-wild style prompts, as illustrated in the first row of Figure 6, the stylized results of DME exhibit noticeable motion distortion. Turning to the results from Motion Puzzle, while the motion style is effectively expressed, the contents are not aligned with the source motion. The result also exhibits motion distortion when using the style motion extract from the video, as shown in the bottom row of Figure 6. In comparison, MotionS stands out for its robustness to arbitrary style cues, demonstrating a more stable ability to generate diverse and stylized results.

***Ablation study.*** We conduct ablative experiments to verify the key designs in our MotionS as Figure 7 shouws, which involves cross-modality style embedding, topology-encoded tokens, and the utilization of graph convolutional layers.

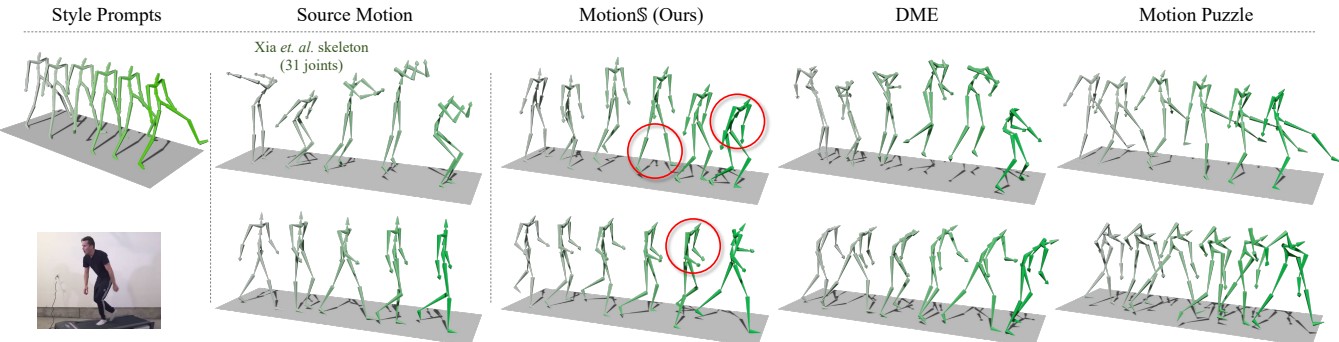

**Figure 6: Qualitative comparison with DME and MotionPuzzle. The results of DME and MotionPuzzle suffer from motion distortion when complex style prompts are used.**

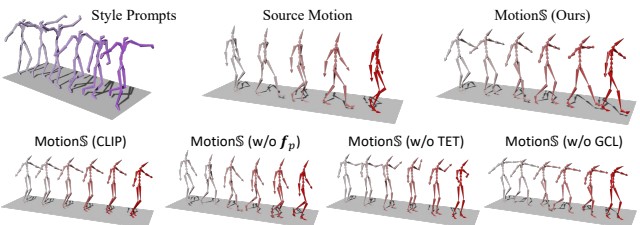

**Figure 7: Qualitative ablation study for key designs.**

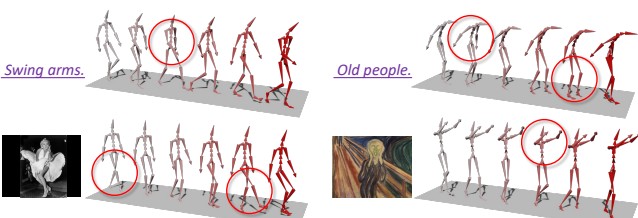

**Figure 8: Evaluation of the generalizability.**

The concept of cross-modality style embedding is inspired by MotionCLIP, with a key distinction being that our approach independently learns the motion content feature and the motion style feature, reconstructing the input motion using both features. To assess this design, we conducted an experiment by employing MotionCLIP as the style encoder and incorporating a learnable canonical decoder in TSD. The results, as shown in Figure 7, indicate that MotionS (CLIP) fails to reconstruct the dynamic source content, yielding a static stylized result. Additionally, in MotionS, we concatenate the temporally repeated style feature with the content feature to obtain the motion feature, as depicted in Figure 3 and Figure 4. This design is crucial for learning dynamic motion styles. The result of MotionS (w/o $f_p$) demonstrates that without this process, the model fails to express the motion style in the generated motion.

The reason for the aforementioned failed result is that the motion style features aligned to the CLIP space only contain static posture information, posing a challenge for the model to transfer dynamic motion styles to the source motion. Thanks to our approach, which involves decoupled learning of motion content and style features, along with the weights transfer strategy of the pre-trained canonical decoder, TSD in our MotionS can be theoretically viewed as comprising two complementary diffusion processes. Initially, it leverages the style feature as a condition to generate the dynamic content of the style motion and subsequently stylizes the source motion in the canonical motion space. Consequently, our cross-modality style embedding plays a pivotal role in achieving flexible motion stylization in the generative process.

On the other hand, topology-encoded tokens and graph convolutional layers have proven effective for skeletal topology shifting. In Figure 7, MotionS (w/o TET) is implemented using the QnA-based network for all modules, maintaining the joint dimension of the features as one. However, this approach disregards the topology differences among various skeletal structures, leading to an inability to accurately express the motion style in the generated results

and causing jetting in the output motion. Furthermore, the result of the model without the GCL is also worse than our MotionS, further confirming that prior adjacency of the skeleton topology can aid the model in perceiving the skeletal structure and achieving cross-skeleton motion style transfer.

***Generalizability.*** MotionS achieves zero-shot style control using unseen style prompts, as demonstrated in Figure 8. The character performs walking motion content stylized by text and images. Despite the implicit nature of the style prompts, such as the text "Old people" and the famous painting "Skrik", MotionS can effectively generate stylized motion that captures the key features of these prompts. This capability is attributed to our cross-modality style embedding, which aligns the style feature with the CLIP space that trained on a tremendous dataset, enhancing the generalizability of MotionS. However, as is common with deep learning-based approaches, the capacity and robustness of the system are constrained by the training data. Thus, MotionS may face limitations when dealing with excessively abstract style descriptions.

***Generative stylization.*** Distinctly differing from existing motion style transfer methods, MotionS functions as a generative pipeline. As illustrated in Figure 9, the source motion depicts the character running in a circle, and MotionS generates stylized results of the character running randomly in any direction with a longer sequence length. With its ability to generate diverse results and accurately express motion styles, MotionS holds significant potential for applications in the field of computer animation.

## 4.2 Quantitative Results

Table 4 presents a quantitative comparison between MotionS and the baselines. It is crucial to emphasize that achieving a balance of good scores across all metrics is preferable to excelling in just a few. For the evaluation on SMPL source motion, we utilize the 100STYLE dataset as style prompts. For the evaluation on Xia source motion, we use the Xia test set from Aberman et al. [2] as style prompts.

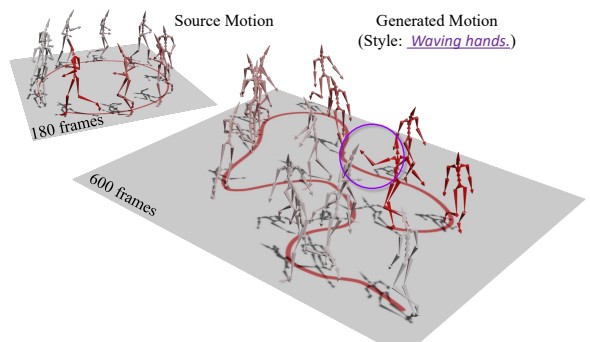

Figure 9: Evaluation of the generative stylization. Motion𝕊 generates diverse motion from a single content sequence.

**Content preservation.** As shown in Table 4, Motion𝕊 effectively preserves the source motion content compared to the baselines. Despite SinMDM* achieving the best FMD, its generated results do not express the motion style, as illustrated in Figure 5. The results of DME and Motion Puzzle reveal lower generalization ability than Motion𝕊, as their performance on Xia source motion is significantly better than on SMPL source motion. However, on both datasets, their performance is worse than Motion𝕊. Additionally, Motion𝕊 (CLIP) and (w/o TET) exhibit worse performance on this metric than ours, further validating the effectiveness of our key designs.

**Style fidelity.** In the evaluation of style fidelity on SMPL source motion, Motion𝕊 achieves the best FMD. On Xia source motion, Motion𝕊 also obtains comparable results with DME and Motion Puzzle. These results demonstrate that Motion𝕊 can effectively stylize the generated motion according to the motion prompts. DME performs best on this metric for Xia source motion, given that we use the data directly from their experiments. However, the representation of motion style through joint positions in DME limits its generalizability to unseen data, leading to the worst result on SMPL source motion. Additionally, Motion𝕊 (CLIP) and (w/o TET) cannot accurately transfer the motion style, as shown in Figure 7, due to their inability to construct the canonical motion space.

**Stylized motion diversity.** Thanks to the diffusion model we employed, Motion𝕊 is capable of generating diverse motion from a single source sequence. The Global Diversity and Local Diversity are calculated based on the source motion. SinMDM obtains the highest values since its generated result is similar to the source motion while lacking distinct style properties. In contrast, the results of SinMDM* are primarily based on the style motion, leading to a deficiency in motion content. On the other hand, the results of Motion𝕊 preserve the source motion content, express style motion features, and exhibit rich diversity.

### 4.3 Additional Evaluation

**Manual adjustment.** As discussed in Section 3.3, the network flow of TSD in our Motion𝕊 is bifurcated into two branches to ensure stable training. Only the stylization branch is utilized during inference to generate the stylized motion. To enhance the flexibility of the pipeline, we introduce a weight parameter $\alpha$, to blend the two branches and perform feature linear interpolation in the canonical motion space. This allows for manual adjustment, providing smooth control over the degree of motion stylization. By gradually scaling $\alpha$, as Figure 10 illustrates, we can obtain results that transition

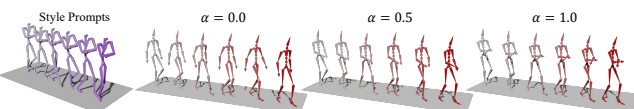

Figure 10: Application of the linear interpolation weight $\alpha$. Manually adjusting $\alpha$ can obtain smooth motion stylization.

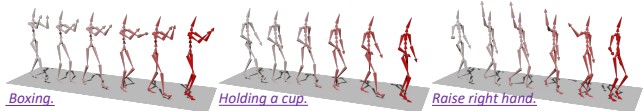

Figure 11: We train the TSD on a text-to-motion dataset and assess our Motion𝕊' performance in motion editing.

smoothly from the source motion to the stylized motion, enabling interactive selection of visually optimal results.

**Editing.** Our Motion𝕊 pipeline can be extended to the application of text-based motion editing. To achieve this, we train the TSD on the BABEL dataset, utilizing it as style motion prompts in alignment with the pre-training step of cross-modality style embedding. As illustrated in Figure 11, we edit the walking motion using various text prompts, and the generated results closely match the given text prompts. However, the huge search space involved in editing motion content for cross-structure skeletons poses challenges for our model to learn. As a result, the visual quality of the editing results is inferior to that of text-based motion generation methods.

## 5 CONCLUSIONS

In this work, we propose a novel generative motion stylization pipeline, Motion𝕊, capable of synthesizing diverse and stylized motion from a single source sequence using multi-modality style descriptions. In Motion𝕊, two key strategies are exploited to embed the cross-modality style prompts and the cross-structure skeleton motion into a canonical motion space. The first strategy is cross-modality style embedding, aligning style motion with the CLIP space and extracting style features from motion, text, or image prompts. The second strategy is cross-structure topology shifting, aligning arbitrary skeleton structures with the SMPL skeleton in latent space, enabling general motion stylization for various characters. In implementation, we construct Motion𝕊 based on the topology-shifted stylization diffusion model and pre-train the style embedding auto-encoder on a large-scale motion, text, and image triplet dataset. Subsequently, we transfer the pre-trained weights of the canonical decoder into the diffusion model to ensure the reconstruction of the dynamic motion style. We conduct extensive experiments to validate the effectiveness of our method, demonstrating that it achieves the state-of-the-art performance compared to the baseline methods.

**Limitations.** One potential drawback lies in the limitation of style descriptions that Motion𝕊 trained on, possibly leading to the generated results that may not accurately express the motion style described in arbitrary custom prompts. Furthermore, foot contact is not our primary focus, it can be addressed using the method proposed in [1]. We acknowledge that some of the generated motion of our Motion𝕊 exhibits motion distortions and artifacts. However, it should be noted that stylizing motion on cross-structure characters using cross-modality sources remains a challenging task that has not been completely solved. We are committed to continuing our efforts to improve the geometric quality of generated motion.

**Acknowledgements.** This work was supported by the Natural Science Fund for Distinguished Young Scholars of Hubei Province No.2022CFA075, the National Natural Science Foundation of China (NSFC) No.62106177, the Fundamental Research Funds for Central Universities No.2042023KF0180.

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
