# OpenReview forum: "Generative Motion Stylization of Cross-structure Characters within Canonical Motion Space"
_acmmm.org/ACMMM/2024/Conference — MM2024 Poster_

### Official Review · Reviewer_kzWd · 2024-04-28

**Rating:** 5
**Confidence:** 3

**Summary:**

1.This paper proposes a novel generative motion stylization pipeline to synthesize cross-structure diverse and stylized motion by utilizing cross-modality style descriptions.
2.The author provides cross-modality style embedding and cross-structure topology shifting, which construct a canonical motion space, enabling skeletal topology perception and flexible style representations for motion stylization.
3.This paper provides extensive experiments to verify the effectiveness and superiority of their method.

**Strengths:**

1.The paper is well writed and easy to follow.
2. The author puts forward a innovative method, which has great pioneering and promoting significance to the field.
3.The author provides extensive experiments which seems good.
4.The idea is not complex but effitive.

**Limitations:**

1. If possible, I hope the author can open the source code for other researchers.
2. Although the result of the experiment is good, I still hope that the author can provide more detailed ablation experiments to verify the role of different techniques in the model.
3. Another point that worries me is the time consumption of the model, although adding more modules may make the results better, whether more time is introduced, and how the model in this paper compares to other models in terms of time consumption is unknown.
4.The compared baseline methods seem not enough, and some evaluation metrics is not clear.
5.The real life usefulness of the method is still an open question, more real-life styles should be verified.

**Suitability:**

3

---

### Official Review · Reviewer_zniN · 2024-05-22

**Rating:** 4
**Confidence:** 3

**Summary:**

This paper proposes  a generative motion stylization pipeline to synthesize diverse and stylized motion. More specifically, cross-modality style prompts are proposed to perceive the cross-structure skeleton topologies, allowing for motion stylization. Extensive qualitative and quantitative comparisons validate the effectiveness of the proposed method.

**Strengths:**

1. The overall content of the paper is well-organized and written, contributing to ease of understanding for readers.
2. Both the quantitative and qualitative results presented in the paper are commendable. The stylized motion modeling is particularly noteworthy, demonstrating a high level of fidelity and realism.

**Limitations:**

1. The paper introduces cross-modality style embeddings as conditional prompts, injected into the latent space via Adaptive Instance Normalization (AdaIN). While certain advancements, similar conditioning strategies have been proposed in previous works [1][2]. To strengthen the paper's contribution, the authors are encouraged to provide more comparative analysis, demonstrating the  improvements of the proposed  method over existing approaches.
[1]  Scalable Diffusion Models with Transformers.
[2]  StyleInject: Parameter Efficient Tuning of Text-to-Image Diffusion Models.

**Suitability:**

3

---

### Official Review · Reviewer_uhXB · 2024-05-24

**Rating:** 4
**Confidence:** 3

**Summary:**

Motivated by existing motion style transfer methods don’t work on cross-skeleton and multi-modality style prompts, the paper proposes a generative motion stylization MotionS. It first encodes the source motion into motion feature by single motion diffusion encoder and into a canonical motion space by canonical decoder. Meanwhile style prompt is encoded into a cross-modality style embedding aligned with CLIP latent space. The style transfer is then performed in the canonical motion space. Finally, the stylized motion is shifted to another topology by diffusion decoder for stylized motion output in different skeletons. Evaluations on 6 motion sequences demonstrate improvements to existing methods.
The key contributions are cross-modality embedding for canonical motion space and cross-structure topology shifting for bridging canonical motion space with specific motion space.

**Strengths:**

- The overall paper is well-written and the motivations/techniques are clearly explained.
- The proposed contributions are technically sound and novel to the motion style transfer field.
- The aspects of evaluation are comprehensive such evaluating the generalizability and generative stylization ability.

**Limitations:**

- The reference [20] claims to support style transfer between different skeletons. The paper is lacking a direct comparison to this existing method.
- Since the evaluation dataset only has 6 motion sequences in total, the quantitative evaluation metrics may be less convincing.
- As the method requires training on every input source motion individually, what are the time to train a model on single sequence?
- When pre-training the cross-modality style embedding, where are the text and image style prompts from?
- Some notations are missing definitions such as N. I can somehow infer that N means the number of joints but it would be better to give definition directly.

**Suitability:**

3

---

### Meta-Review · Area_Chair_knxi · 2024-06-29

**Recommendation:** Accept (Poster)
**Confidence:** 4

**Metareview:**

Pros: The paper reads well. The proposed idea seems interesting. Good results are reported, both qualitatively and quantitative.

Cons: Very limited empirical evaluations (with 6 motion sequences). The authors are encouraged to empirically verify on a larger number of (and also broader range of) motions. Also need to empirically compare and demonstrate its results with those SOTA ones e.g. [1].